# Arrhythmic Manifestations of Cardiac Amyloidosis: Challenges in Risk Stratification and Clinical Management

**DOI:** 10.3390/jcm12072581

**Published:** 2023-03-29

**Authors:** Natallia Laptseva, Valentina A. Rossi, Isabella Sudano, Rahel Schwotzer, Frank Ruschitzka, Andreas J. Flammer, Firat Duru

**Affiliations:** 1Division of Heart Failure, Clinic for Cardiology, University Heart Center Zurich, 8091 Zurich, Switzerland; 2Amyloidosis Network Zurich, University Hospital Zurich, 8091 Zurich, Switzerland; 3Clinic for Medical Oncology and Hematology, University Hospital Zurich, 8091 Zurich, Switzerland; 4Division of Arrhythmia and Electrophysiology, Clinic for Cardiology, University Heart Center Zurich, 8091 Zurich, Switzerland; 5Center for Integrative Human Physiology, University of Zurich, 8091 Zurich, Switzerland

**Keywords:** cardiac amyloidosis, arrhythmia, heart failure, cardiomyopathy, sudden death

## Abstract

Amyloidosis is a systemic disease characterized by extracellular deposits of insoluble amyloid in various tissues and organs. Cardiac amyloidosis is a frequent feature of the disease, causing a progressive, restrictive type of cardiomyopathy, and is associated with adverse clinical outcomes and increased mortality. The typical clinical presentation in patients with cardiac amyloidosis is heart failure (HF) with preserved ejection fraction. Most patients present with typical symptoms and signs of HF, such as exertional dyspnea, pretibial edema, pleural effusions and angina pectoris due to microcirculatory dysfunction. However, patients may also frequently encounter various arrhythmias, such as atrioventricular nodal block, atrial fibrillation and ventricular tachyarrhythmias. The management of arrhythmias in cardiac amyloidosis patients with drugs and devices is often a clinical challenge. Moreover, predictors of life-threatening arrhythmic events are not well defined. This review intends to give a deepened insight into the arrhythmic features of cardiac amyloidosis by discussing the pathogenesis of these arrhythmias, addressing the challenges in risk stratification and strategies for management in these patients.

## 1. Introduction

Amyloidosis is a systemic disease that is characterized by extracellular deposits of an insoluble fibrillar protein called *amyloid* within various tissues and organs, mainly the heart, kidneys and the peripheral nervous system [1]. There are more than 30 amyloidogenic precursor proteins in humans. The clinical phenotype depends on the nature of the amyloid and stage of disease, and can vary considerably in different patients. Cardiac amyloidosis is a frequent feature of the disease (in up to 60% of patients), causing a progressive, restrictive type of cardiomyopathy, and is associated with adverse clinical outcomes [2,3].

There are two main forms of amyloidosis that significantly affect the heart: *light chain amyloidosis* (AL) and *transthyretin amyloidosis* (ATTR) [4,5]. The two forms have a completely different pathophysiology and their treatment differs considerably [6,7]. AL type is caused by overproduction and misfolding of antibody light chain fragments by an underlying clonal plasma cell in the bone marrow. Other vital organs may also be involved in AL amyloidosis, such as kidneys, liver, peripheral and autonomic nervous system. The majority of the patients are male and over the age of 50, although the disease may also occur in younger patients. Early diagnosis of AL cardiac amyloidosis is very important since the disease, if untreated, typically progresses rapidly and causes severe heart failure (HF), life-threatening arrhythmias, and results in death. ATTR, which occurs due to deposits of misfolded monomers of transthyretin (a small molecule mainly produced by the liver) can be observed either as a genetic disease (variant ATTR, or vATTR) or more commonly as a nonhereditary disease, called wild-type ATTR (wtATTR). In general, progression of ATTR cardiac amyloidosis (especially wtATTR) is slower as compared to that observed in AL cardiac amyloidosis, and untreated affected individuals may live many years after initial manifestation of disease.

The typical clinical presentation in patients with cardiac amyloidosis is HF with preserved ejection fraction (HFpEF). Most patients present with typical symptoms and signs of HF, such as exertional dyspnea, pretibial edema, pleural effusions and angina pectoris due to microcirculatory dysfunction. These symptoms occur due to the infiltrative, restrictive nature of the amyloid deposits, particularly as a result of the rapid rise of filling pressures during diastole. However, patients may also frequently encounter various bradyarrhythmias due to conduction system disease, such as atrioventricular (AV) nodal block, and tachyarrhythmias, such as atrial fibrillation and ventricular arrhythmias, which may potentially result in life-threatening consequences [8].

The purpose of this review is to give a deepened insight into the arrhythmic features of cardiac amyloidosis by discussing the pathogenesis of these arrhythmias, addressing the challenges in risk stratification and strategies for management in patients suffering from this disease. 

## 2. Arrhythmic Manifestations of Cardiac Amyloidosis

The common arrhythmia characteristics in patients with cardiac amyloidosis are summarized in Table 1. The occurrence of various arrhythmias and the potential role for device therapy in different forms of cardiac amyloidosis are shown in Table 2. The prognostic factors known to date in the development of brady- and tachyarrhythmias in cardiac amyloidosis are listed in Table 3. The main pathophysiologic factors causing arrhythmias in cardiac amyloidosis are demonstrated in Figure 1.

## 3. Conduction System Disturbances

A somewhat underappreciated manifestation common to all forms of cardiac amyloidosis is conduction system disease. The spectrum of conduction system disturbances is broad and may include sinus nodal disease, intra- or interatrial abnormality, AV nodal or infranodal dysfunction, bundle branch blocks, intraventricular conductions disturbances, and combinations of the above (Figure 2). In addition, surface ECG during sinus rhythm may show other typical features of cardiac amyloidosis, such as low voltages on extremity leads (Figure 3). Sick sinus disease may be the initial presentation of isolated cardiac involvement in AL [9,10]. Moreover, advanced interatrial block is common in patients with cardiac amyloidosis and has adverse impact on atrial function [11]. However, AV conduction disturbances are more prevalent in cardiac amyloidosis, as reflected by first and higher degrees of AV block on surface ECG and long HV intervals in electrophysiologic recordings [12]. Widening of QRS complexes, suggesting intraventricular conduction disturbances, are also common in cardiac amyloidosis. The later finding is in fact more profound in ATTR than in AL [13]. Likewise, bundle branch blocks are commonly observed in cardiac amyloidosis. Right bundle branch block is more prevalent because the thin and slender shape of the right bundle makes it more vulnerable to amyloid deposits, as compared to the left bundle [13].

There are multiple possible factors that may be involved in the pathogenesis of conduction system disturbances in cardiac amyloidosis: (1) The thickening of myocardium due to amyloid deposition may disrupt electrical impulse propagation along conduction fibers; (2) Possible cytotoxic effects of certain amyloid precursor proteins (e.g., AL variants) may induce oxidative stress and apoptosis, and interfere with intracardiac conduction; (3) Neurotoxic amyloid deposition within interstitial space and consequent sympathetic denervation may contribute to conduction system disturbances; (4) ATTR may interfere with cytoplasmic calcium signaling [14].

### Pacemaker Therapy in Cardiac Amyloidosis

Patients with cardiac amyloidosis who develop conduction system disturbances often require implantation of a pacemaker. In a cohort of 405 AL patients, Porcari et al. reported 8.9% pacemaker implantation rate within 3 years of diagnosis [15]. In this cohort, PR interval of >200 ms and QRS duration of >120 ms on surface ECG and history of atrial fibrillation predicted a future pacemaker implantation. However, there seems to be a distinction with respect to pacemaker indication in different types of cardiac amyloidosis. Thirty percent of patients with wtATTR and 15% of patients with vATTR already had a pacemaker implanted at the time of diagnosis compared to only 1% of patients with AL, suggesting a different electrophysiological pathophysiology between the ATTR and AL types [16]. However, this discrepancy may also be explained by differences in severity of the diseases. AL type is typically a more aggressive and a rapidly progressive disease. In contrast, slow progression of ATTR, associated with lack of symptoms at early disease, may explain why some patients with this disease may receive pacemaker implantation even before the underlying disease is diagnosed. On the other hand, there is no indication for prophylactic implantation of a pacemaker in these patients. Based on the available evidence, it may be advisable to screen for bradycardia and AV block in 24 h Holter recordings at 6-month intervals [15].

Given the progressive nature of cardiac amyloidosis, percentage of RV pacing typically increases during the course of disease. Rehorn et al. reported that most patients eventually became totally pacemaker-dependent at around 5 years after implantation [17]. A higher RV pacing burden is associated with adverse remodeling and worsening of HF in patients with cardiac amyloidosis. Donnellan et al. demonstrated worsening of left ventricular ejection fraction (LVEF), worsening in New York Heart Association (NYHA) functional class and worsening of mitral regurgitation in patients with ATTR cardiac amyloidosis, if RV pacing was more than 40% [18]. In contrast, significant improvements in LVEF, NYHA functional class, and mitral regurgitation severity occurred in those with cardiac resynchronization therapy (CRT). Therefore, the authors suggested preferential use of CRT in patients with ATTR cardiac amyloidosis who have a high RV pacing burden. It is important to note that indication for CRT implantation was based on established guidelines and the vast majority of the patients (23 out of 25) had left bundle branch block before implantation in this cohort. Moreover, 20 patients (80%) were biventricular paced >99% of the time. Conduction system pacing (with placement of the RV lead in the His-bundle region or at a high septal location) may also theoretically help to minimize depression of ventricular function due to ventricular dyssynchrony from RV pacing and be beneficial in pacing with cardiac amyloidosis. However, there is a need for clinical data to understand the true impact of CRT (or conduction system pacing) on clinical outcomes in cardiac amyloidosis.

## 4. Supraventricular Tachyarrhythmias

Patients with cardiac amyloidosis may experience elevated resting heart rates. Due to the restrictive nature of amyloid cardiomyopathy, the elevated filling pressures lead to low end-diastolic volume and reduced stroke volume. The systolic blood pressure is reduced, especially during exercise. Moreover, these patients often have orthostasis due to peripheral polyneuropathy. In this constellation, compensatory elevation of the heart rate is the only mechanism to sustain the cardiac output in these patients. For this reason, most patients with cardiac amyloidosis do not tolerate beta-blockers.

Atrial tachyarrhythmias are commonly observed in patients with all forms of cardiac amyloidosis. In particular, atrial fibrillation is present in up to 70% of patients at the time of diagnosis (Figure 4) [14]. The high prevalence of atrial arrhythmias is possibly due multiple proarrhythmogenic factors: (1) Elevated intra-atrial pressures; (2) Left atrial dilatation; (3) Infiltration of amyloid fibrils within the atria; (4) Typical older age of patients with ATTR cardiac amyloidosis. From an electrophysiological stand of view, elevated intra-atrial pressures may increase ectopic beats arising from the pulmonary veins and trigger initiation of atrial fibrillation. On the other hand, patchy islands of amyloid infiltration in the atria may facilitate microreentrant circuits and perpetuate the occurrence of atrial tachyarrhythmias.

One important aspect of atrial fibrillation in patients with cardiac amyloidosis is the more common occurrence of intracardiac thrombi and thromboembolic complications with this disease, as compared to those observed with other types of cardiomyopathies. An autopsy study published by Feng et al. demonstrated presence of intracardiac thrombi in approximately one-third of patients who died due to cardiac amyloidosis [19], and a similar incidence was reported using transthoracic echocardiography in a subsequent study from the same group [20]. Thromboembolic risk in cardiac amyloidosis patients is very high and thrombi were detected in the left atrium even without atrial fibrillation [21]. AL type cardiac amyloidosis was demonstrated to be associated with more thromboembolic complications, probably due to more impaired LVEF in this disease. Patients with ATTR cardiac amyloidosis may be at a comparably lower risk for intracardiac thrombus formation, possibly due to preserved EF, but even in these patients, there seems to be no association between CHA_2_DS_2_-VASc scores and thrombus formation within the left atrial appendage [21]. For the above-mentioned reasons, anticoagulation must be initiated in all cardiac amyloidosis patients with atrial fibrillation. Whether anticoagulation is reasonable in cardiac amyloidosis patients with high filling pressures and immobile atria, even without documented atrial fibrillation, is still a matter of debate [22]. It may be advisable to screen for atrial fibrillation with Holter ECGs periodically, for example at 6-month intervals.

The presence of atrial fibrillation may put patients with cardiac amyloidosis at risk for developing cardiogenic shock, possibly due to lack of atrial contribution to impaired left ventricular filling in cardiac amyloidosis, particularly during rapid heart rates [23]. On the other hand, in contrast to patients with most other cardiomyopathies, cardiac amyloidosis patients typically have controlled ventricular rates during atrial fibrillation due to concomitant occurrence of conduction system disease.

Due to restrictive ventricular filling and reduced stroke volume, the use of antiarrhythmic drugs in patients with cardiac amyloidosis is often a clinical challenge [24]. Rapid heart rates that decrease diastolic filling times are often not well tolerated. On the other hand, medications that reduce the heart rate, and in particular beta-blockers, are poorly tolerated due to repression of the compensatory increase in heart rate and resultant further reduction of the cardiac output. Likewise, calcium antagonists are considered harmful in cardiac amyloidosis patients [25,26,27]. Nevertheless, at low doses, they may be considered for their antihypertensive action during early disease. The use of digoxin in cardiac amyloidosis is often discouraged due to its profound binding to amyloid, which may intensify its pharmacological effect [24,28]. However, available data in the literature does not support the perception of proarrhythmogenesis with the use of digoxin [29]. Moreover, recent studies suggest that digoxin may be cautiously used for rate control at low doses in selected patients, while closely monitoring drug levels, renal function and electrolytes [30,31].

Among the antiarrhythmic drugs, amiodarone is often the drug of choice for patients with cardiac amyloidosis because it has the most favorable short-term safety profile and is relatively well tolerated [32,33]. The role of catheter ablation for atrial fibrillation in patients with cardiac amyloidosis is limited as high rates of recurrence after pulmonary vein isolation have been reported [34].

## 5. Ventricular Tachyarrhythmias

Ventricular arrhythmias are commonly observed in patients with cardiac amyloidosis, which range from ventricular premature beats and nonsustained VT to sustained VT and ventricular fibrillation (Figure 5). Sudden cardiac death (SCD) may account for half of cardiac deaths in cardiac amyloidosis [35]. Although electromechanical dissociation due to pulseless electrical activity (PEA) is considered to be the most common cause of SCD in cardiac amyloidosis, ventricular tachyarrhythmias may also result in SCD in these patients [36].

A number of risk factors are known to be associated with higher overall mortality in cardiac amyloidosis patients. In AL amyloidosis, several staging models such as the Mayo and Boston University models were introduced to identify patients who are at high risk [37,38]. Likewise, staging systems, which include biomarkers like NTproBNP (N-terminal prohormone of brain natriuretic peptide) and high sensitivity cTnT (cardiac Troponin T), or decreased estimated glomerular filtration rate (eGFR) were proposed for ATTR cardiac amyloidosis [39,40]. Despite the above-mentioned efforts to predict overall mortality, it remains to be a challenge to identify patients who are at risk for SCD, and particularly, SCD due to ventricular tachyarrhythmias. For example, there seems to be no correlation between cardiac biomarkers levels and the risk for ventricular tachyarrhythmia occurrence [41].

As it is the case for the etiopathogenesis of conduction disorders and atrial tachyarhythmias, the arrhythmogenic substrate for the occurrence of ventricular arrhythmias in cardiac amyloidosis is also due to amyloid infiltration in extracellular spaces and impairment of cardiomyocyte function. Arrhythmias associated with amyloid infiltration were shown to correlate with the extent of conduction tissue infiltration, as demonstrated in LV endomyocardial biopsy specimens [42].

Myocardial scarring and fibrosis that can easily be detected by late gadolinium enhancement (LGE) during cardiac magnetic resonance imaging (CMR), being either global subendocardial or transmural, or in some cases with a patchy pattern, may be considered as surrogate markers for ventricular proarrhythmogenesis [43,44]. For example, in patients with hypertrophic cardiomyopathy, fibrosis is an established component of a prognostic score in predicting life-threatening tachyarrhythmias [43]. In the case of cardiac amyloidosis, however, the extent of LGE-uptake is a more controversial parameter to predict the arrhythmia risk, because LGE uptake in cardiac amyloidosis is not only caused by scar/fibrosis, but also by the deposition of the amyloid itself. Moreover, amyloid deposition may also have other electrophysiological characteristics in causing arrhythmias, in comparison to the typical reentry caused by a fibrous scar.

The electrophysiological substrate in patients with cardiac amyloidosis is complex and includes signal fractionation, slow intraventricular conduction, low epicardial signal amplitudes, prolonged and spatially dispersed repolarization, and were more prominent in patients with AL amyloidosis [45]. Marked prolongation of HV interval in AL cardiac amyloidosis, which indicates severe disease of the distal His-Purkinje system, was determined to be the only independent predictor for SCD [35]. Inducibility of monomorphic VT was rare in cardiac amyloidosis and non-inducibility showed little prognostic value, as this is also the case in other non-ischemic cardiomyopathies [35].

### Implantable Cardioverter-Defibrillators in Cardiac Amyloidosis

Identification of patients with cardiac amyloidosis who may benefit from implantable cardioverter-defibrillator (ICD) therapy is a challenge. It remains unclear whether ICD prevents SCD in these patients. A combined analysis of 11 retrospective studies studying the outcomes of 720 ICD patients with a variety of cardiac amyloidosis etiologies showed that 23% of these patients received appropriate therapies and 88% of them survived the arrhythmic events following ICD interventions [35]. However, in seven of these studies, the survival rate for patients who received appropriate therapies was only 22%, suggesting that ICD therapy had probably no or very modest effect on survival. The frequent occurrence of electromechanical dissociation is at least partially responsible for the low efficacy of device therapy in cardiac amyloidosis patients [46]. The incidence of inappropriate ICD interventions was rather low (7%) in the entire cohort [35]. A more recently published study reported high rates of ventricular arrhythmias and appropriate ICD therapies among a unique cohort of largely hereditary ATTR patients with a high rate of systolic HF [47].

Nonsustained VT is common in patients with cardiac amyloidosis but its role in SCD prediction seems to be limited [48]. However, it may still be considered as a risk marker in certain patients, for example in the early stage of AL. The current Heart Rhythm Society (HRS) Consensus Statement gives a Class IIb recommendation for ICD implantation for primary prevention in patients with AL amyloidosis and non-sustained VT, if the expected survival is longer than one year [49]. Likewise, syncope is also common in cardiac amyloidosis and seems to have a limited role in SCD risk prediction. It is rather nonspecific because it can occur due to conduction disturbances and, possibly more commonly, as a result of hypotension due to orthostasis or diuretic therapy.

The classical indication for ICD implantation for primary prevention in patients with most cardiomyopathies is LV dysfunction with an EF ≤ 35%. A similar approach for decision-making does not apply to cardiac amyloidosis because the decline in LVEF occurs during later course of the disease. This makes the role of other imaging modalities other than conventional echocardiography (such as speckle tracking imaging showing reduced LV global longitudinal strain and CMR demonstrating the presence of transmural LGE) particularly important in detecting the extent of disease early on, and possibly identifying those who may benefit from prophylactic ICD implantation [50].

The histopathological basis of LGE in patients with cardiac amyloidosis has not been well studied. Since amyloid deposition may also lead to global subendocardial LGE update, it may be important to identify the presence of real scar/fibrosis, which is a well-known risk predictor for ventricular tachyarrhythmias. Most studies on LGE apply the null-point method, in which the signal from the least-enhancing region (accepted as the *normal* myocardium) is measured, which is used as the reference. In patients with cardiac amyloidosis, however, LGE images show dynamic changes upon the elapsed time after contrast injection [51]. Hashimura et al. reported that LGE acquired at the mid-diastolic phase using the “fixed inversion time method” corresponded to interstitial amyloid deposition and subendocardial fibrosis that was most likely attributable to microvascular obstruction [52].

Patients with cardiac amyloidosis who already have an LVEF ≤ 35% are more likely to develop electromechanical dissociation as terminal arrhythmic event, and hence, less likely to benefit from ICD therapy [41,53]. The same also applies to those patients who have very high NT-proBNP levels [53]. The 2022 ESC Guidelines for the Management of Patients with Ventricular Arrhythmias and the Prevention of SCD give a class IIa indication for ICD use in patients with AL or ATTR cardiac amyloidosis who have a hemodynamically not-tolerated VT after careful discussion of the competing risks of non-arrhythmic death and non-cardiac death [54].

## 6. Conclusions

Patients with cardiac amyloidosis typically present with HFpEF but may also frequently encounter various arrhythmias. Bradyarrhythmias due to conduction system disturbances such as sinus and AV nodal disease are not rare and may necessitate the implantation of a pacemaker. Likewise, tachyarrhythmias are rather common in cardiac amyloidosis. Atrial fibrillation is the most common arrhythmia and anticoagulation should be considered in all patients. Ventricular tachyarrhythmias may occur, putting the patients at risk for life-threatening consequences. The management of arrhythmias in patients with cardiac amyloidosis with drugs and devices is often a clinical challenge. Electromechanical dissociation due to PEA is considered to be the most common cause of SCD, and therefore, the role of ICD in patients for primary prevention is controversial and possibly limited. Moreover, predictors of ventricular arrhythmia risk are not well defined. Apart from the fibrotic scar, amyloid deposition may cause LGE uptake in CMR depending on stage and type of amyloidosis but its direct impact on arrhythmogenesis has not been studied. Future studies with large sample sizes and well-defined patient populations are required to determine the true impact of various arrhythmias and the role of therapeutic interventions during the clinical course of cardiac amyloidosis.

## Figures and Tables

**Figure 1 jcm-12-02581-f001:**
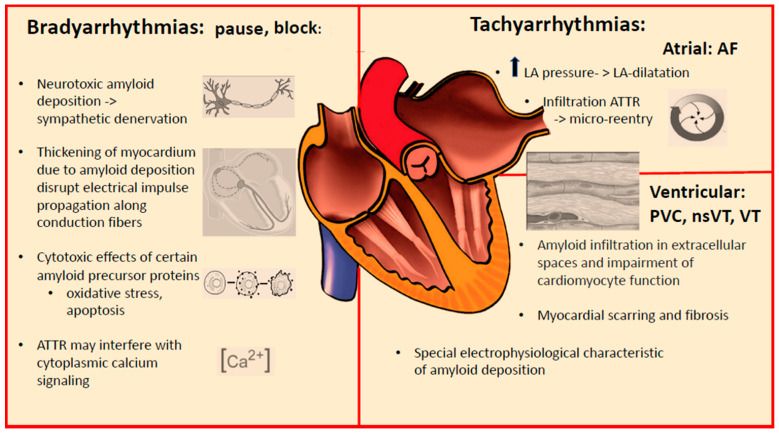
Main pathophysiologic factors causing arrhythmias in cardiac amyloidosis.LA: left atrium; AF: atrial fibrilation; PVC: premature ventricular contraction; nsVT: nonnonsustained ventricular tachycardia; VT: ventricular tachycardia; arrow means elevated.

**Figure 2 jcm-12-02581-f002:**
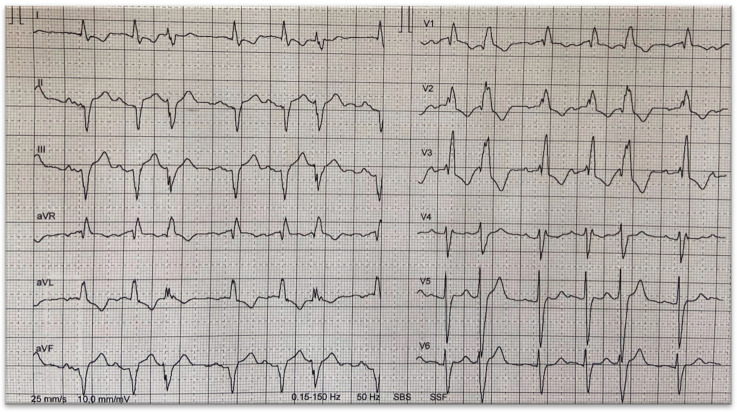
12-lead surface ECG of a 53-year-old male patient with wtATTR demonstrating 1st degree AV block, left anterior hemiblock, right bundle branch block and supraventricular ectopic beats.

**Figure 3 jcm-12-02581-f003:**
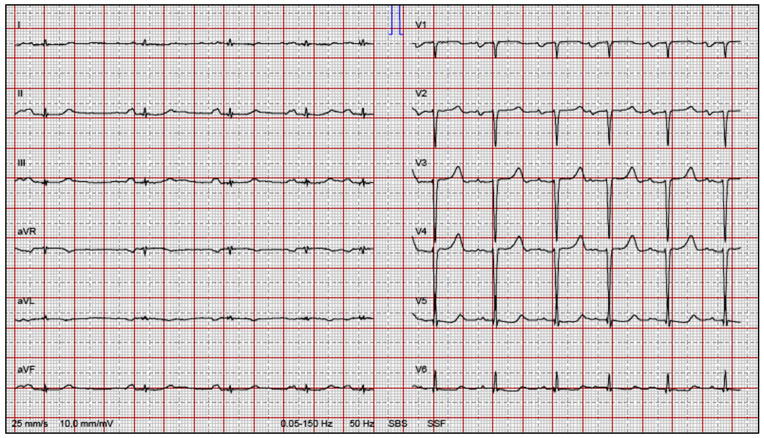
12-lead surface ECG of a 77-year-old male patient with AL, demonstrating low voltages on extremity leads, left atrial abnormality as well as poor R-wave progression on precordial leads.

**Figure 4 jcm-12-02581-f004:**
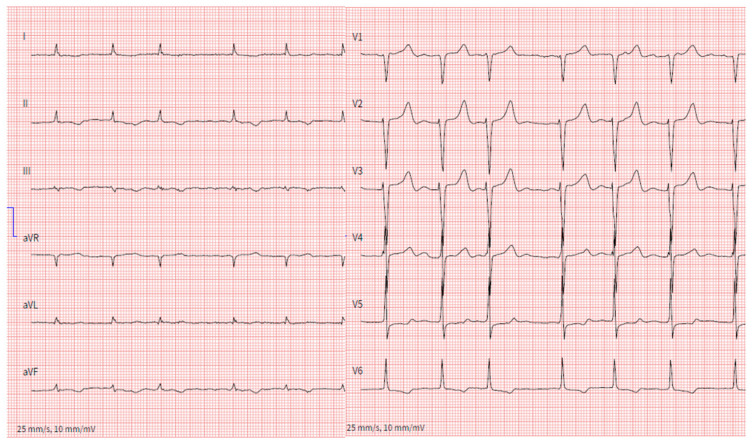
12-lead surface ECG of a 73-year-old male patient with wtATTR, low voltages on extremity leads and atrial fibrillation with a mean heart of 75 bpm.

**Figure 5 jcm-12-02581-f005:**
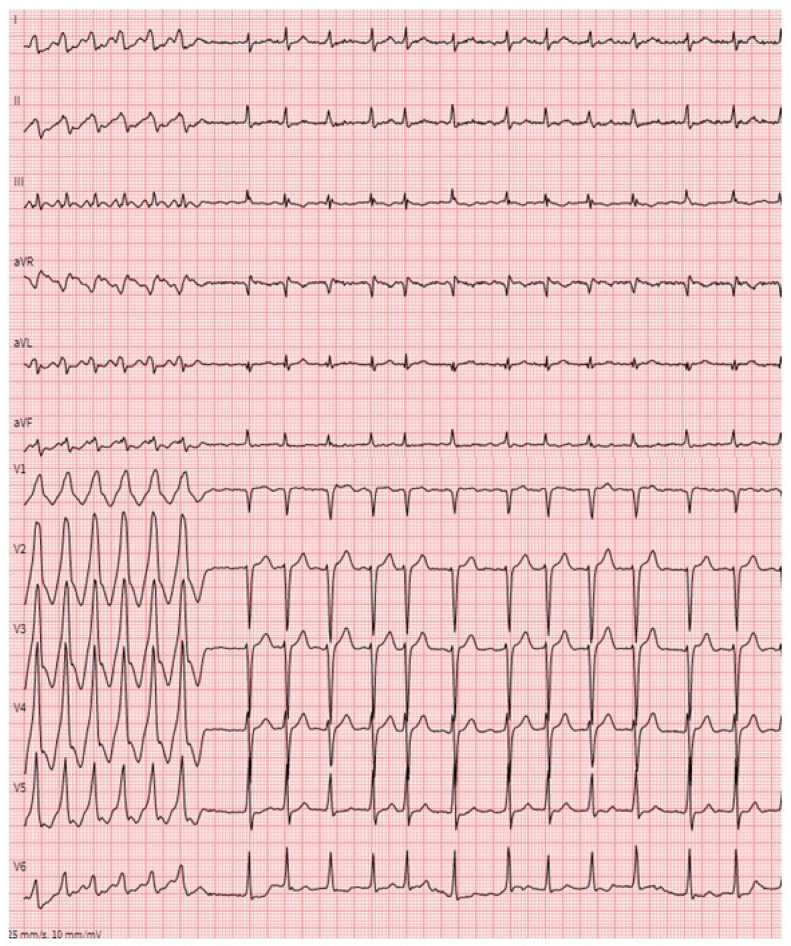
12-lead surface ECG of a 67-year-old male patient with wtATTR showing a monomorphic VT, which spontaneously terminates into atrial fibrillation.

**Table 1 jcm-12-02581-t001:** Typical arrhythmia characteristics in patients with cardiac amyloidosis.

AF is common and is often poorly tolerated due to impaired ventricular filling.
Anticoagulation should be considered in all patients with AF.
AV conduction during AF may be slower due to concomitant AV conduction disease.
Management of AF with antiarrhythmics or ablation is often challenging.
Nonsustained VT may have limited prognostic value in predicting SCD.
Syncope may occur due to multiple mechanisms and does not predict SCD.
Monomorphic VT is rarely inducible during electrophysiologic study.
Pulseless electrical activity is a more common cause of SCD than VT/VF or asystole.
The role of ICD use in primary prevention of SCD is yet unclear.

AV: atrioventricular; AF: atrial fibrillation; VT: ventricular tachycardia; SCD: sudden cardiac death; VF: ventricular fibrillation; ICD: implantable cardioverter defibrillator

**Table 2 jcm-12-02581-t002:** Arrhythmia risk and indication for device therapy in various forms of cardiac amyloidosis.

	AL	vATTR	wtATTR
**Arrhythmia risk**			
Sinus node disease	+/++	++	++
AV conduction disease	++	+++	+++
Atrial fibrillation	++/+++	+++	+++
nsVT/VT	+++	++/+++	++/+++
SCD risk	+++	++	++
**Device indication**			
PM	+	+++	++/+++
CRT	+/++	++/+++	++
ICD (primary prevention)	+/?	+/?	+/?

+ demonstrates degree of increased arrhythmia risk or degree of device indication (+, ++ or +++); +/++ means between + and ++; ? means controversial; AL: light chain amyloidosis; vATTR: variant transthyretin amyloidosis; wtATTR: wild-type transthyretin amyloidosis; AV: atrioventricular; PM: pacemaker; CRT: cardiac resynchronization therapy; nsVT: nonsustained ventricular tachycardia; SCD: sudden cardiac death; ICD: implantable cardioverter defibrillator.

**Table 3 jcm-12-02581-t003:** Prognostic factors in the development of clinically relevant brady- and tachyarrhythmias in cardiac amyloidosis.

**Bradyarrhythmias** Potential candidates who may require cardiac pacing * − PR interval >200 ms − QRS duration >120 ms − History of atrial fibrillation − Long HV intervals in electrophysiologic testing − High degree AV block − ATTR cardiac amyloidosis
**Tachyarrhythmias** Patients at risk for atrial fibrillation − Left atrial dilatation − Advanced interatrial block − Old age Potential candidates who may require ICD therapy * − Increased NTproBNP and high sensitivity cTnT − Decreased eGFR − LGE uptake in CMR (role uncertain) − nsVT (role controversial and possibly limited) − Syncope (role controversial and possibly limited)

* The list does not include conventional indications for pacemaker and ICD therapy. HV: His-Ventricle interval; AV: atrioventricular; ATTR: transthyretin amyloidosis; ICD: implantable cardioverter defibrillator; NTproBNP: N-terminal prohormone of brain natriuretic peptide; cTnT: cardiac Troponin T; eGFR: estimated glomerular filtration rate; LGE: late gadolinium enhancement: CMR: cardiac magnetic resonance imaging; nsVT: nonsustained ventricular tachycardia.

## Data Availability

Not applicable.

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
