# Peer review of "Arrhythmic Manifestations of Cardiac Amyloidosis: Challenges in Risk Stratification and Clinical Management"

_jcm, 2023, doi:10.3390/jcm12072581_

Round 1

Reviewer 1 Report

1.       It is my absolute pleasure to review the manuscript " Arrhythmic Manifestations of Cardiac Amyloidosis: Challenges in Risk Stratification and Clinical Management". Comments as listed below.

2.       This review is well-written with abundant summarization of the current evidence. There are statements and minor syntax need to be reconfirmed.

3.       “Drugs and devices is often a clinical challenge”, however, the current review seems to left out the part of challenge in medication. While the contents and the illustrations are very representative, the authors might consider reorganizing to mention this part or to mainly focus on the non-drug management.

4.      There are some undefined acronyms in the main text. E.g.: NTproBNP, cMRI/CMR.

5.       As the standout point of the review, I think the Tables might need further organization:
- Table 1 is only listed the highlight characteristics, which I did not see any “compared to those with cardiomyopathies causing HFrEF” and the supporting references. These contents can be integrated into the table to provide a more informative review.

- In Table 2, what is the “/” & “?” refer? The current format could not fulfill self-explanatory.

Reviewer 2 Report

The authors provided a comprehensive review about arrhythmic manifestation of cardiac amyloidosis properly debating current challenges in their management. However, the role of medical therapy, particularly antiarrhythmic drugs, have not been adequately discussed. Given the variety of pathogenetic factors involved in the arrhythmic substrate, is there a significant role for conventional antiarrhythmic drugs in prevention of atrial fibrillation recurrency? Are there some significant differences compared to general population? And what about management of ventricular arrhythmias. Are there some evidences that conventional drugs, e.g. class I and class III, are safe and effective in this patient population? Are there differences compared to general population? Please the authors should address this issue.

A comprehensive figure, perhaps through the use of images or small icons, summarizing the main pathophysiologic factors responsible for arrhythmias may be useful.

What a clinician should consider in prognostic stratification of the arrhythmic burden of these patients has been well addressed but sometimes seems dispersive. A summary table of prognostic factors highlighted to date in the development of supraventricular and ventricular arrhythmias would be helpful.

Round 2

Reviewer 1 Report

The current review version has used the Track Change Mode with a good amount of syntax revision. The contents overall should be fit, however, the text must be thoroughly reviewed.

Author Response

At the suggestion of the Reviewer, we have thoroughly reviewed the entire manuscript for English language, syntax and spell checks, and now provide a revision with fine/minor corrections. We believe that we have adequately addressed the Reviewer's comment and the revised manuscript is now suitable for publication in the Journal of Clinical Medicine.